# A Novel Diagnosis Scheme against Collusive False Data Injection Attack

**DOI:** 10.3390/s23135943

**Published:** 2023-06-26

**Authors:** Jiamin Hu, Xiaofan Yang, Luxing Yang

**Affiliations:** 1Department of School of Big Data & Software Engineering, Chongqing University, Chongqing 400044, China; jiaminhu@cqu.edu.cn; 2College of Information Technology, Deakin University, Melbourne, VIC 3125, Australia; y.luxing@deakin.edu.au

**Keywords:** wireless sensor network, collusive false data injection attack, diagnosis scheme, watchdog, autoregressive moving average model, principal component analysis, diagnosis algorithm

## Abstract

The collusive false data injection attack (CFDIA) is a false data injection attack (FIDA), in which false data are injected in a coordinated manner into some adjacent pairs of captured nodes of an attacked wireless sensor network (WSN). As a result, the defense of WSN against a CFDIA is much more difficult than defense against ordinary FDIA. This paper is devoted to identifying the compromised sensors of a well-behaved WSN under a CFDIA. By establishing a model for predicting the reading of a sensor and employing the principal component analysis (PCA) technique, we establish a criterion for judging whether an adjacent pair of sensors are consistent in terms of their readings. Inspired by the system-level fault diagnosis, we introduce a set of watchdogs into a WSN as comparators between adjacent pairs of sensors of the WSN, and we propose an algorithm for diagnosing the WSN based on the collection of the consistency outcomes. Simulation results show that the proposed diagnosis scheme achieves a higher probability of correct diagnosis.

## 1. Introduction

Wireless sensor networks (WSNs) are networks of wirelessly interconnected sensor nodes that collect data about the surrounding environment [1]. With the rapid popularization of Internet of Things (IoT) applications, WSNs have penetrated nearly all aspects of human life, ranging from industry and transportation to healthcare and military affairs [2]. Typically, sensors have limited energy, limited memory storage, and limited computing/communication capabilities, and are deployed in unattended and abominable environments. As a result, WSNs are vulnerable to a variety of cyber attacks. Consequently, the security of WSNs has received considerable attention from the network security community [3,4,5,6].

### 1.1. Problem Formulation

False data injection attacks (FDIAs) are cyber attacks on WSNs where false data are stealthily injected into the physically captured sensors [7,8]. FDIAs would render the compromised sensors to deliver wrong data to the base station. As a result, the decision-maker at the base station would make an incorrect decision, leading to serious consequences.

An attacker can obtain key information from a compromised sensor to gain control over it, which leads to a chance that proactive security mechanisms are useless in detecting FDIAs. Therefore, the best way to counteract FDIAs is detection by analyzing the measurements themselves. The spatiotemporal correlation of inter-measurements is a solution used in many studies to detect FDIAs [9,10,11]. Due to the continuity of physical phenomena, the measurements of each sensor are temporally correlative in time. Due to the high-density network topology of WSNs, the inter-measurements of adjacent sensors are spatially correlative. When compromised and genuine sensors coexist, the inconsistency of measurements will lead to correlation failure.

However, attackers aim to minimize the risk of being detected by employing resourceful and sophisticated strategies. Most previous works on FDIA detection have been focused on situations where independent false data are injected into different captured sensors [10,12,13,14,15,16]. In this paper, we consider FDIAs in which the readings of some adjacent pairs of compromised sensors are modified in a coordinated manner so that the false readings still look spatially–temporally correlated. We refer to such FDIAs as collusive FDIAs or simply *CFDIAs*. As a result, the conventional methods for detecting an ordinary FDIA fail when used to detect a CFDIA. Consequently, it is crucial to investigate the following problem:

*CFDIA diagnosis problem*: Identify the compromised sensor nodes under a CFDIA in a WSN with no natural anomalies.

Inspired by the system-level fault diagnosis, a solution based on hybrid detection will be created for the CFDIA diagnosis problem. To our knowledge, this is the first time such an attempt has been made.

### 1.2. Main Contributions

Our main contributions are sketched below.

We define a new kind of false data injection attack to WSNs, i.e., a conclusive false data injection attack (CFDIA), and we propose a new problem (i.e., the CFDIA diagnosis problem) aiming to identify the compromised sensors in a WSN under a CFDIA.We establish an autoregressive moving average (ARMA) model for predicting the current reading of a sensor using its historical readings. Based on the prediction model and by employing the principal component analysis (PCA) technique, we establish a model for judging if an adjacent pair of sensors are consistent in terms of their readings.Inspired by the system-level fault diagnosis, we introduce a set of watchdogs in the WSN under CFDIA as comparators between adjacent pairs of sensors within their respective communication range. These watchdogs deliver their respective collections of consistency outcomes to the base station. The base station collects all the received consistency outcomes to form a complete syndrome.We design an algorithm for identifying the abnormal sensors based on the complete syndrome. Through extensive simulation experiments, we conclude that the diagnosis algorithm achieves a higher probability of correct diagnosis.

The subsequent materials are organized in this fashion: In Section 2, the related works are reviewed. In Section 3, some terms, notations, and assumptions are introduced. The diagnosis scheme is described in detail in Section 4, and the effectiveness of the diagnosis scheme is corroborated through simulation experiments in Section 5. Finally, this work is summarized by Section 6.

## 2. Related Work

In this section, we make comments on the previous work that is related to the present paper, aiming to highlight the novelty of our work.

### 2.1. System-Level Fault Diagnosis

The system-level fault diagnosis aims to identify faulty units in a computer system based on a collection of test/comparison outcomes between adjacent units [17,18]. There are two different system-level diagnosis approaches: the test-based diagnosis and comparison-based diagnosis [17].

In the test-based diagnosis, for each adjacent pair of units, one unit serves as the tester and the other as the testee; the tester assigns a computational job to the testee, the testee performs the computation and returns the computational result to the tester, and the tester compares the result with its own result and judges the testee to be fault-free or faulty according to whether the two results are identical or not. Finally, a diagnosis algorithm is performed based on the collection of test outcomes [17,19].

In the comparison-based diagnosis, each adjacent pair of units perform the same computation, and the computational results are compared. That the two results are identical implies that the two units are either both fault-free or faulty. On the contrary, that the two results are not identical implies that at least one of the two units is faulty. Finally, a diagnosis algorithm is performed based on the collection of comparison outcomes [18,20].

The system-level fault diagnosis has been applied to the detection of faulty sensors in WSNs [21,22,23,24]. Inspired by the system-level fault diagnosis, an algorithm for diagnosing the abnormal sensors in the WSN under a CFDIA based on the collection of the consistency outcomes is proposed without considering the existence of natural anomalies, and validates the effectiveness of the diagnosis algorithm through simulation experiments. To our knowledge, however, the system-level fault diagnosis has not been applied to the identification of compromised sensor nodes in WSNs under an FDIA, let alone the identification of captured sensors under a CFDIA.

### 2.2. FDIAs Detection of WSNs

FDIAs are known for their severe impact and are one of the widely studied cyber-attacks in smart grids, power systems, and WSNs [10,14]. In the research area of the FDIAs detection problem of WSNs, many studies focus on exploring techniques from the sensor measurements.

Ref. [25] proposed a generalized distributed anomaly detection scheme based on the spatio-temporal correlation of physical processes against FDIAs in WSNs. Ref. [26] presented a method combining a measurement check and authentication strategies to detect FDIAs in WSNs. Ref. [27] exploited the sensor data correlation in time and space to identify the falsified data in the industrial Internet-of-Things. Ref. [28] utilized the observed spatio-temporal and multivariate-attribute sensor correlations to detect FDIAs in WSNs. Ref. [29] addressed the issue of detecting FDIAs based on spatial correlation to dynamic WSNs. Ref. [30] suggested a method using temporal, spatial, and event-based correlations to prevent FDIAs in WSNs.

Few works cover detecting CFDIAs in WSNs. Ref. [31] revealed that wireless nodes usually have some correlation patterns in communication metrics, which can be used to defend against CFDIAs in WSNs. Ref. [32] proposed a wavelet transform method based on wavelet transform to detect CFDIAs. Ref. [33] exploited the spatio-temporal correlation of heterogeneous sensor data to detect CFDIAs in low-density WSNs. Since the detection of CFDIAs requires comparing measurements over a broader set of sensors, these efforts are based on a centralized detection scheme to provide a global view. However, a compromised node not only injects false data into itself, but it may also inject false data into routed packets, potentially leading to a higher false positive rate.

In this paper, we propose a hybrid detection scheme. We define watchdog as a kind of expensive dedicated hardware device that is deployed in the area monitored by the WSN in concern, is within the communication range of the base station serving the WSN, and can acquire the readings of the sensors that are within the communication range of the device. A set of watchdogs are deployed in the area for judging the consistency of adjacent pairs of sensors in terms of the spatial–temporal correlation of their readings. All the consistency outcomes are delivered directly to the base station. To our knowledge, a hybrid detection scheme has not yet been applied to the CFDIA diagnosis problem.

## 3. Preliminary Knowledge

### 3.1. WSNs with Watchdogs

Consider a WSN used for gathering a kind of environmental data within a specific area. Let $R_{1}$ denote the communication radius of the base station serving the WSN. Let $R_{2}$ denote the common communication radius of the sensors in the WSN. Let the undirected graph G=(V,E) denote the topological structure of the WSN, i.e., $V=\{v_{1},\cdots ,v_{N}\}$ stands for the set of sensors in the WSN, and $\{v_{i},v_{j}\}\in E$ if and only if $v_{i}$ is within the communication range of $v_{j}$. Let $r_{i}\left(t\right)$ denote the measurement reading of the node $v_{i}$ at time *t*.

Define watchdog as a kind of expensive dedicated hardware device that is deployed in the area monitored by the WSN, is within the communication range of the base station, and can acquire the readings of the sensors that are within the communication range of the device. Suppose a set of watchdogs are deployed within the monitored area. Let $R_{3}$ denote the common communication radius of the watchdogs. Suppose $R_{2}<R_{3}<R_{1}$. Let $W=\{w_{1},\cdots ,w_{M}\}$ denote the set of the watchdogs. Suppose the watchdogs are used for periodically acquiring the measurement readings of the sensors within their respective communication ranges. Let S={1,2,⋯,K} denote the set of time points at which the readings of the sensors are acquired by their respective neighboring watchdogs.

### 3.2. Collusive False Data Injection Attack

A collusive false data injection attack (CFDIA) is a false data injection attack in which the readings of some adjacent pairs of compromised sensors are modified in a coordinated manner so that the changed readings are still spatially–temporally correlated. Owing to the attacker’s limited budget, assume (i) the compromised nodes are less than normal nodes, and (ii) the compromised sensors are concentrated in a small area. Figure 1 illustrates a CFDIA to a toy WSN of seven nodes.

### 3.3. Autoregressive Moving Average Models

An autoregressive model builds on the assumption that there is a linear relationship between the current value of a variable and its own historical values. A moving average model assumes that the current value of a variable depends not only on the current information but also on previous information. The model obtained by combining an autoregressive model with a moving average model is referred to as an autoregressive moving average (ARMA) model [34]. An ARMA model of order (p,q) is formulated as follows.
(1)$$r\left(t\right)=\mu +\sum_{l=1}^{p}\phi_{l}r\left(t-l\right)+\sum_{l=1}^{q}\psi_{l}\epsilon \left(t-l\right)+\epsilon \left(t\right).$$

Here, r(t) stands for the value of the variable *r* at time *t*; μ, $\phi_{l}$, and $\psi_{l}$ are model parameters; and ϵ(t) stands for the value of the independent error at time *t*, which follows a Gaussian distribution with zero mean.

### 3.4. Principal Component Analysis

The principal component analysis (PCA) is a commonly used technique for reducing the dimensionality of large datasets and increasing data interpretability. The PCA creates new uncorrelated variables (the principal components) by solving an eigenvalue/eigenvector problem [35]. The PCA has been applied to FDIA detection [36,37].

## 4. A Diagnosis Scheme against CFDIA

In this section, we propose a diagnosis scheme against a CFDIA. The scheme consists of two phases: the syndrome generation phase and the CFDIA diagnosis phase, which are stated as follows.

*Phase I: Syndrome generation*. In this phase, each watchdog collects a set of readings of the sensors monitored by the watchdog and conducts a spatio-temporal correlation analysis between each adjacent pair of sensors, forming a partial syndrome. All the watchdogs deliver their own partial syndromes directly to the base station. A (complete) syndrome is generated by merging the partial syndromes.*Phase II: CFDIA Diagnosis*. Taking the syndrome as input, perform an algorithm for diagnosing a CFDIA. As a result, the compromised nodes are identified.

Next, let us discuss the two phases in detail.

### 4.1. Syndrome Generation

#### 4.1.1. Consistency Criterion

The syndrome on a WSN under a CFDIA refers to the collection of the consistency outcomes between adjacent pairs of nodes of the WSN. The syndrome is the basis of CFDIA diagnosis. The key to generating the syndrome is to establish a consistency criterion. For this purpose, we need to discuss temporal correlation and spatial correlation between adjacent pairs of nodes, respectively.

First, there is a temporal correlation of each node in terms of their readings. Let ${\tilde{r}}_{i}\left(t\right)$ denote the predicted value of $r_{i}\left(t\right)$. We assume that for i=1,⋯,n and for all *t*, ${\tilde{r}}_{i}\left(t\right)$ obeys the following ARMA model of order (p,q):(2)$${\tilde{r}}_{i}\left(t\right)=\mu_{i}+\sum_{l=1}^{p}\phi_{l}^{i}r_{i}\left(t-l\right)+\sum_{l=1}^{q}\psi_{l}^{i}\epsilon_{i}\left(t-l\right)+\epsilon_{i}\left(t\right).$$

The parameters in the model can be estimated using the historical data.

**Remark 1.** 
*ARMA is used to model linear relationships, which is suitable for stationary stochastic processes. However, the presence of seasonality and trends in the time-series sensor readings may introduce nonlinear non-stationary sequences. Therefore, the time-series sensor readings need to be pre-processed before extracting the spatio-temporal correlation of adjacent nodes. There is a need for stationary identification. If the time-series sensor readings are not stationary, we can employ the differencing method on the time-series sensor readings to remove seasonality and trends.*


Second, there is a spatial correlation between each adjacent pair of nodes in terms of their readings. We use the PCA to reveal the spatial correlation. Let ${\bar{r}}_{i}$(resp. ${\bar{r}}_{j}$) denote the mean of historical readings of $v_{i}$(resp. $v_{j}$). The correlation coefficient of the predicted readings of an adjacent pair of nodes, $v_{i}$ and $v_{j}$, is calculated as follows.
(3)$$\delta_{ij}=\frac{\sum_{t=1}^{K}\left({\tilde{r}}_{i}\left(t\right)-{\bar{r}}_{i}\right)\left({\tilde{r}}_{j}\left(t\right)-{\bar{r}}_{j}\right)}{K-1}.$$

The covariance matrix of the predicted readings of $v_{i}$ and $v_{j}$ reads
(4)$$\Lambda_{ij}=\left[\begin{array}{cc} \delta_{ii} & \delta_{ij}\\ \delta_{ji} & \delta_{jj} \end{array}\right].$$

Let $\lambda_{ij}^{1}$ (resp. $\lambda_{ij}^{2}$) denote the largest (resp. second largest) eigenvalue of the matrix $\Lambda_{ij}$. Let ${\vec{\mu }}_{ij}^{1}$ (resp. ${\vec{\mu }}_{ij}^{2}$) denote the unit eigenvector of $\Lambda_{ij}$ associated with $\lambda_{ij}^{1}$ (resp. $\lambda_{ij}^{2}$).
(5)$$\Lambda_{ij}{\vec{\mu }}_{ij}^{1}=\lambda_{ij}^{1}{\vec{\mu }}_{ij}^{1},\;\Lambda_{ij}{\vec{\mu }}_{ij}^{2}=\lambda_{ij}^{2}{\vec{\mu }}_{ij}^{2}.$$

Assume ${\vec{\mu }}_{ij}^{1}$ and ${\vec{\mu }}_{ij}^{2}$ are linearly independent. Then, ${\vec{\mu }}_{ij}^{1}$ is orthogonal to ${\vec{\mu }}_{ij}^{2}$.

The consistency ellipse at time *t* with the confidence degree 1−θ (here, θ≤0.1), denoted as $\Gamma_{ij}^{1-\theta }\left(t\right)$, can be calculated by taking $\left({\tilde{r}}_{i}\left(t\right),{\tilde{r}}_{j}\left(t\right)\right)$ as its center, taking ${\vec{\mu }}_{ij}^{1}$ (resp. ${\vec{\mu }}_{ij}^{2}$) as the direction of its major axis (resp. minor axis), taking $\lambda_{ij}^{1}$ (resp. $\lambda_{ij}^{2}$) as the ratio of its long radius (resp. its short radius), and choosing the confidence degree 1−θ. Let ${\bar{\Gamma }}_{ij}^{1-\theta }\left(t\right)$ denote the closed region surrounded by $\Gamma_{ij}^{1-\theta }\left(t\right)$. Then, ${\bar{\Gamma }}_{ij}^{1-\theta }\left(t\right)$ is the confidence region with the confidence degree 1−θ. Consequently, we present the following:

*Consistency criterion*: An adjacent pair of nodes, $v_{i}$ and $v_{j}$, are consistent at time *t* with the confidence degree 1−θ if $\left({\tilde{r}}_{i}\left(t\right),{\tilde{r}}_{j}\left(t\right)\right)\in {\bar{\Gamma }}_{ij}^{1-\theta }\left(t\right)$. Otherwise, they are inconsistent with the confidence degree 1−θ.

See Figure 2a for a schematic explanation of the consistency criterion. For brevity, we remove “at time *t*” and “with confidence degree 1−θ” in the criterion.

#### 4.1.2. Syndrome and Partial Syndrome

Let σ(u,v)=0 (resp. 1) denote that the adjacent pair of nodes, *u* and *v*, are consistent (resp. inconsistent). We refer to the collection
(6)σ={σ(u,v):{u,v}∈E}
as the *syndrome* on the WSN. For each adjacent pair of nodes, *u* and *v*, we make the following reasonable assumptions:If *u* and *v* are both normal, then σ(u,v)=0 with probability 1−θ.If one of *u* and *v* is normal and the other is abnormal, then σ(u,v)=1 with probability 1−θ.If *u* and *v* are both abnormal, then σ(u,v)=0 or 1.

For each watchdog $w_{m}$, let $V_{m}$ denote the set of nodes that are monitored by $w_{m}$. We refer to the collection
(7)$$\sigma_{m}=\left\{\sigma \left(u,v\right):u,v\in V_{m},\left\{u,v\right\}\in E\right\}$$
as the *partial syndrome* associated with $w_{m}$.

Each watchdog can acquire the partial syndrome associated with it. All partial syndromes can be delivered by the watchdogs to the base station. Finally, the syndrome can be generated at the base station by merging the received partial syndromes.

### 4.2. CFDIA Diagnosis

We intend to identify all the abnormal nodes of a WSN by interpreting the syndrome. For this purpose, we need to introduce the following terms and notations.

**Definition 1.** 
*Let p be the a priori probability of a node of WSN being compromised.*


**Definition 2.** 
*Let σ be a syndrome on the WSN G=(V,E), e∈E. The edge e is referred to as a 0-edge or a 1-edge according to σ(e) = 0 or 1.*


**Definition 3.** 
*Let σ be a syndrome on the WSN G=(V,E). The 0-subgraph of G is defined as a subgraph of G, denoted $G_{0}=\left(V,E_{0}\right)$, such that $E_{0}$ is the set of 0-edges of G.*


**Definition 4.** 
*Let σ be a syndrome on the WSN G=(V,E), $G_{0}=\left(V,E_{0}\right)$ the 0-subgraph of G. Let $H=\{H_{i}=\left(U_{i},E_{i}\right):1\leq i\leq r\}$ be the collection of connected components of $G_{0}$. The 1-condensed graph of G is defined as a graph $G^{*}=\left(U^{*},E^{*}\right)$ such that (i) $U^{*}=\left\{u_{1}^{*},\cdots ,u_{r}^{*}\right\}$, (ii) $\left\{u_{i}^{*},u_{j}^{*}\right\}\in E^{*}$ if and only if there is a 1-edge of G that connects a node in $U_{i}$ with a node in $U_{j}$.*


Based on the previously introduced assumptions about the relationship between the states of two adjacent nodes and their consistency, we have the following results.

**Theorem 1.** 
*Let σ be a syndrome on the WSN G=(V,E), e={u,v}∈E. If p<<0.5, the following assertions hold true:*
*1.* 
*σ(u,v)=0 implies u and v are either both normal with a higher probability (w.h.p.) or both abnormal w.h.p.*
*2.* 
*σ(u,v)=1 implies at least one of u and v is abnormal w.h.p.*



The following theorem is a direct consequence of this theorem.

**Theorem 2.** 
*Let σ be a syndrome on the WSN G=(V,E). Let $G_{0}=\left(V,E_{0}\right)$ be the 0-subgraph of G. Let $H=\{H_{i}=\left(U_{i},E_{i}\right):1\leq i\leq r\}$ be the set of connected components of $G_{0}$. If p<<0.5, the following assertions hold true:*
*1.* 
*For 1≤i≤r, either (i) the nodes in $U_{i}$ are all normal w.h.p., or (ii) the nodes in $U_{i}$ are all abnormal w.h.p.*
*2.* 
*If there is a 1-edge connecting $U_{i}$ with $U_{j}$, then either (i) the nodes in $U_{i}$ are all normal and the nodes in $U_{j}$ are all abnormal w.h.p, or (ii) the nodes in $U_{j}$ are all abnormal and the nodes in $U_{j}$ are all normal w.h.p.*



Based on the theorem, we present an algorithm (i.e., the CFDIA algorithm given in Algorithm 1) for identifying the abnormal nodes in a WSN under a CFDIA. The correctness of the algorithm is guaranteed by the following observation.

**Algorithm 1** CFDIA-DIAG.      **Input**: A WSN G=(V,E) under CFDIA, a syndrome σ on *G*.   **Output**: A subset $V_{a}\subseteq V$ that is diagnosed to be the set of abnormal nodes.
1:Find the 0-subgraph of *G*, denoted $G_{0}=\left(V,E_{0}\right)$;2:Find all the connected components of $G_{0}$, denoted $H_{i}=\left(U_{i},E_{i}\right)$, i=1,⋯,r;3:Find the 1-condensed graph of *G*, denoted $G^{*}=\left(U^{*},E^{*}\right)$, where $U^{*}=\left\{u_{1}^{*},\cdots ,u_{r}^{*}\right\}$;4:// label all nodes of $G^{*}$ with *Z*; //5:**for** each $u^{*}\in U^{*}$ **do**6:  $l(u^{*})\leftarrow Z$;7:// label all nodes of $G^{*}$ with *X* or *Y* through depth-first search; //8:Let *Q* be an empty queue;9:Choose a node $u^{*}$ with maximum degree from $U^{*}$;10:$l(u^{*})\leftarrow X$; $Q\leftarrow Q+u^{*}$;11:**while** $G^{*}$ has a node with label *Z* **do**12:  fetch node $u^{*}$ with the most nodes from *Q*; $Q\leftarrow Q-u^{*}$;13:  **for** each $v^{*}\in U^{*}$ that is adjacent to $u^{*}$ **do**14:    **if** 
$l(v^{*})=Z$ 
**then**15:      $Q\leftarrow Q+v^{*}$;16:      **if** 
$l(u^{*})=X$ 
**then**17:       $l(v^{*})\leftarrow Y$;18:      **else**19:       $l(v^{*})\leftarrow X$;20:// determine $V_{a}$; //21:**if** 
$|{⋃}_{l(u_{i}^{*})=X}U_{i}\left|\leq \right|{⋃}_{l(u_{i}^{*})=Y}U_{i}|$ 
**then**22:  $V_{a}\leftarrow {⋃}_{l(u_{i}^{*})=X}U_{i}$;23:
**else**
24:  $V_{a}\leftarrow {⋃}_{l(u_{i}^{*})=Y}U_{i}$;  **return** $V_{a}$.


**Theorem 3.** 
*Consider a connected WSN under a CFDIA. If p<<0.5, then the CFDIA algorithm identifies the abnormal nodes correctly w.h.p.*


As the time overhead of the CFDIA-DIAG algorithm is dominated by the O(|V|+|E|) time needed to perform the search-first search in the algorithm, we obtain that the worst-case time complexity of the diagnosis algorithm is O(|V|+|E|). Additionally, the space complexity of the diagnosis algorithm is O(|V|+|E|) as well.

## 5. Effectiveness of the Proposed Diagnosis Algorithm

This section is devoted to investigating the effectiveness of the CFDIA-DIAG algorithm through simulation experiments.

### 5.1. Metrics of Effectiveness of a Diagnosis Algorithm

In order to measure the effectiveness of the CFDIA algorithm, below let us introduce a pair of metrics of effectiveness of a diagnosis algorithm.

**Definition 5.** 
*Let G=(V,E) be a WSN under a CFDIA, A be the set of abnormal nodes of G, and σ be a syndrome produced by A. Let DIAG be a diagnosis algorithm. Let B be the set of nodes that are diagnosed to be abnormal by running DIAG on (G,σ).*
*1.* 
*The diagnosis accuracy of DIAG with respect to (w.r.t.) (G,A,σ) is defined as*

(8)
$$DA_{DIAG}\left(G,A,\sigma \right)=\frac{\left|B\right|}{\left|A\right|}.$$

*2.* 
*The false positive rate of DIAG w.r.t. (G,A,σ) is defined as*

(9)
$$FPR_{DIAG}\left(G,A,\sigma \right)=\frac{\left|B⋂(V-A)\right|}{\left|V-A\right|}.$$

*3.* 
*The false negative rate of DIAG w.r.t. (G,A,σ) is defined as*

(10)
$$FNR_{DIAG}\left(G,A,\sigma \right)=\frac{\left|A⋂(V-B)\right|}{\left|A\right|}.$$




### 5.2. Experiment Preparation

First, consider two additional diagnosis algorithms. The first one is almost the same as the CFDIA-DIAG algorithm, with the only exception of the sentence in line 10 of the CFDIA-DIAG algorithm being replaced with the sentence “arbitrarily choose a node $u^{*}$ from $U^{*}$”. We refer to this diagnosis algorithm as the *Random-Search algorithm*. The second is based on the *Correlation-Voting* solution proposed in Ref. [33].

Second, consider two different kinds of FDIAs: the simple FDIA (SFDIA) and the CFDIA. For the former, the readings of the compromised sensors are all enhanced by a larger fraction. For the latter, the readings of each adjacent pair of compromised sensors are changed in a coordinated manner.

Third, consider three synthetic WSNs, denoted $G_{1}$, $G_{2}$, and $G_{3}$, of sensor nodes that are with a communication radius of 20 m and are placed randomly in a square region of size 120×120 m^2^, $G_{1}$; $G_{2}$, and $G_{3}$ have 50 nodes, 100 nodes, and 150 nodes, respectively. For each normal node $v_{i}$ and any time *t*, assume $r_{i}\left(t\right)$ follows the Gaussian distribution $G(\mu_{i},\sigma^{2})$, where $\mu_{i}\in \left\{10,15,20\right\}$, $\sigma^{2}=1$, and the correlation coefficient of the readings of each adjacent pair of sensors is 0.9. Suppose a set of nine watchdogs with a communication radius of 40 m are deployed regularly in the region. See Figure 3 for the distribution of $V_{1}$ and nine watchdogs.

Fourth, consider a real-world WSN, $G_{4}$, of 45 effective sensors used for gathering the environmental PM2.5 data, which is located in Krakow, Poland [38]. Here, the common sensing rate of the sensors is one reading per hour, and the average degree of the WSN is 21.16.

### 5.3. Experiments and Analysis of Experimental Results

**Experiment 1.** 
*Consider the WSN $G_{2}$. Let p∈P={0.03,0.06,⋯,0.3}. Let $A_{p}$ be a set of abnormal nodes randomly produced based on p. Let $\sigma_{p}^{c}$ be the syndrome produced by $A_{p}$ under the CFDIA that the readings of each of the compromised nodes are enhanced or reduced by 10%, and $\sigma_{p}^{s}$ the syndrome produced by $A_{p}$ under the SFDIA that the readings of all compromised are enhanced by 40%.*
*1.* 
*For each p∈P, running CFDIA-DIAG, Random-Search, and Correlation-Voting on $\left(G_{2},\sigma_{p}^{c}\right)$, we obtain their DA, FPR, and FNR, which are shown in Figure 4a–c. It is seen that (i) the diagnosis accuracy of CFDIA-DIAG is higher than those of the other two algorithms, and (ii) the false positive rate and false negative rate of CFDIA-DIAG is lower than those of the other two algorithms. Hence, we conclude that CFDIA-DIAG is more effective than the other two algorithms in the CFDIA situation.*
*2.* 
*For each p∈P, running CFDIA-DIAG, Random-Search, and Correlation-Voting on $\left(G_{2},\sigma_{p}^{s}\right)$, we obtain their DA, FPR, and FNR, which are shown in Figure 4d–f. It is seen that (i) the diagnosis accuracy of CFDIA-DIAG is higher than those of the other two algorithms, and (ii) the false positive rate and false negative rate of CFDIA-DIAG is lower than those of the other two algorithms. Hence, we conclude that CFDIA-DIAG is more effective than the other two algorithms in the SFDIA situation.*



**Experiment 2.** 
*Consider the three WSNs: $G_{1}$, $G_{2}$, and $G_{3}$. Let p∈P={0.03,0.06,⋯,0.3}. Let $A_{p,i}$ be a set of abnormal nodes of $G_{i}$ randomly produced based on p. Let $\sigma_{p,i}^{c}$ be the syndrome on $G_{i}$ produced by $A_{p,i}$ under the CFDIA that the readings of each of the compromised nodes are enhanced or reduced by 10%; $\sigma_{p,i}^{s}$ the syndrome on $G_{i}$ produced by $A_{p,i}$ under the SFDIA that the readings of all the compromised are enhanced by 40%.*
*1.* 
*For each p∈P and each i∈{1,2,3}, running CFDIA-DIAG on $\left(G_{i},\sigma_{p,i}^{c}\right)$, we obtain its DA, FPR, and FNR, which are shown in Figure 5a–c. It is seen that (i) the diagnosis accuracy of CFDIA-DIAG is higher when run on denser WSNs, and (ii) the false positive rate and false negative rate of CFDIA-DIAG is lower when run on denser WSNs. Hence, we conclude that in the CFDIA situation, CFDIA-DIAG is more effective when run on dense WSNs.*
*2.* 
*For each p∈P and each i∈{1,2,3}, running CFDIA-DIAG on $\left(G_{i},\sigma_{p,i}^{s}\right)$, we obtain its DA, FPR, and FNR, which are shown in Figure 5d–f. It is seen that (i) the diagnosis accuracy of CFDIA-DIAG is higher when run on denser WSNs, and (ii) the false positive rate and false negative rate of CFDIA-DIAG is lower when run on denser WSNs. Hence, we conclude that in the SFDIA situation, CFDIA-DIAG is more effective when run on dense WSNs.*



**Experiment 3.** 
*Consider the WSN $G_{4}$. The real-world time-series sensor readings are stationary by first difference. Let p∈P={0.03,0.06,⋯,0.3}. Let $A_{p}$ be a set of abnormal nodes randomly produced based on p. Let $\sigma_{p}^{c}$ be the syndrome produced by $A_{p}$ under the CFDIA that the current reading of each of the compromised nodes is replaced with its largest reading in the past K−1 time points, and $\sigma_{p}^{s}$ the syndrome produced by $A_{p}$ under the SFDIA that the readings of all compromised are enhanced by 100%.*
*1.* 
*For each p∈P, running CFDIA-DIAG, Random-Search, and Correlation-Voting on $\left(G_{4},\sigma_{p}^{c}\right)$, we obtain their DA, FPR, and FNR, which are shown in Figure 6a–c. It is seen that (i) the diagnosis accuracy of CFDIA-DIAG is higher than those of the other two algorithms, and (ii) the false positive rate and false negative rate of CFDIA-DIAG is lower than those of the other two algorithms. Again, we conclude that CFDIA-DIAG is more effective than the other two algorithms in the CFDIA situation.*
*2.* 
*For each p∈P, running CFDIA-DIAG, Random-Search, and Correlation-Voting on $\left(G_{4},\sigma_{p}^{s}\right)$, we obtain their DA, FPR, and FNR, which are shown in Figure 6d–f. It is seen that (i) the diagnosis accuracy of CFDIA-DIAG is higher than those of the other two algorithms, and (ii) the false positive rate and false negative rate of CFDIA-DIAG is lower than those of the other two algorithms. Additionally, we conclude that CFDIA-DIAG is more effective than the other two algorithms in the SFDIA situation.*



## 6. Conclusions and Future Work

A novel diagnosis scheme against a conclusive false data injection attack (CFDIA) has been proposed. First, a set of special watchdogs are deployed in a WSN under a CFDIA to collect consistency outcomes of adjacent pairs of sensor nodes and to deliver them to the base station, forming a syndrome. Second, inspired by the system-level fault diagnosis, a CFDIA diagnosis algorithm is presented. The effectiveness of the algorithm is corroborated through simulation experiments. By executing the diagnosis algorithm on the syndrome received by the base station, the set of compromised nodes is identified correctly with a higher probability.

According to the above three metrics of effectiveness, the method proposed in this paper is better than the compared method, but there are some limitations. Firstly, this paper considers only scenarios where attacks exist, and it is not yet designed to distinguish between malicious attacks and natural anomalies deviating from wide-sense stationary jointly Gaussian processes, including faults, disruptive events, and major disruptions. Secondly, the system-level fault diagnosis should be generalized to probabilistic system-level diagnoses to improve on the diagnosis accuracy and reduce the false positive rate and false negative rate of our proposed diagnosis algorithm [39]. Therefore, in future works, we should further optimize the method. Additionally, the proposed diagnosis algorithm should be extended to the diagnosis of mobile networks under a CFDIA [40,41]. Watchdogs can be made mobile to strike a balance between diagnosis accuracy and energy consumption, and the work can be done in the framework of game theory [42,43]. Finally, the methodology developed in the present paper may be applied to some other cybersecurity issues such as defense against advanced persistent threats [44,45].

## Figures and Tables

**Figure 1 sensors-23-05943-f001:**
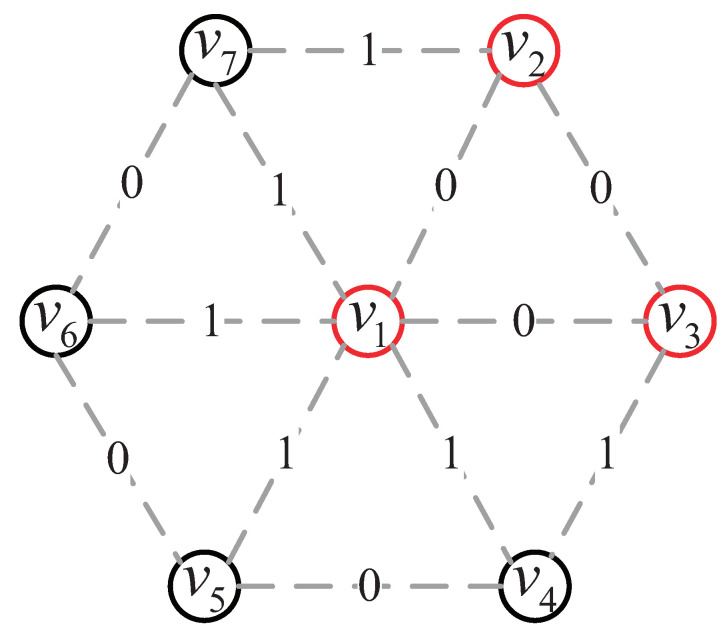
A CFDIA against a toy WSN, where each black circle (resp. red circle) represents a normal node (resp. compromised node), each edge represents the two associated nodes which are within each other’s communication range, and “0” (resp. “1”) represents that there is (resp. there is no) a spatial–temporal correlation between the two associated nodes.

**Figure 2 sensors-23-05943-f002:**
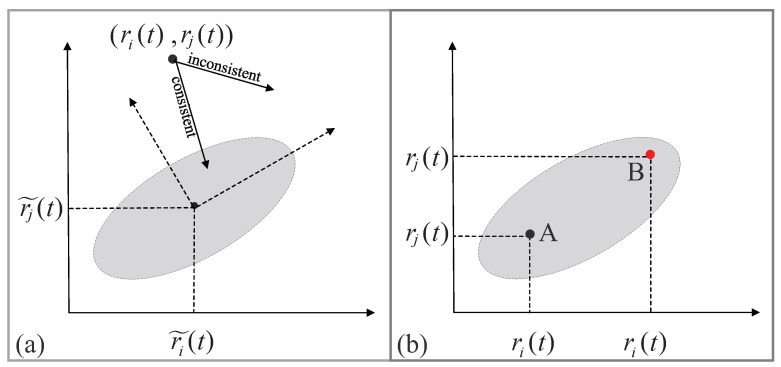
(**a**) The confidence region ${\bar{\Gamma }}_{ij}^{1-\theta }\left(t\right)$ with the confidence degree 1−θ. (**b**) A glance of a CFDIA, where the point *A* represents the values of $r_{i}\left(t\right)$ and $r_{j}\left(t\right)$, and the point *B* represents the false values of $r_{i}\left(t\right)$ and $r_{j}\left(t\right)$.

**Figure 3 sensors-23-05943-f003:**
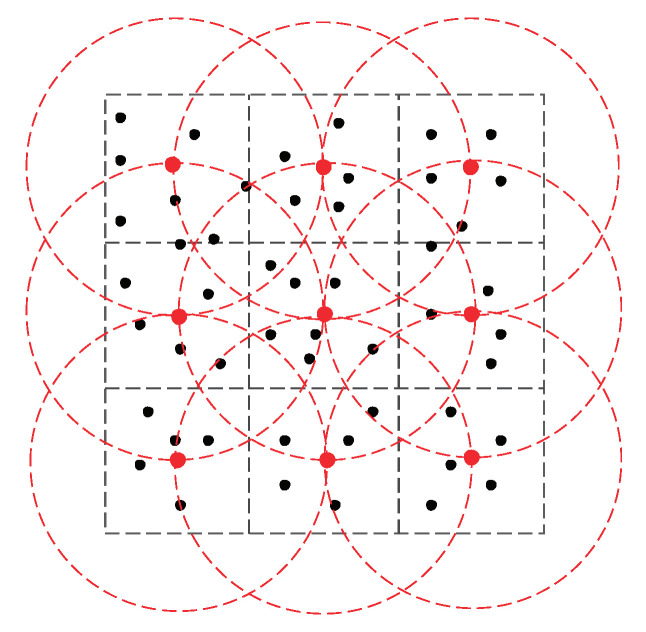
Distribution of $V_{1}$ and 9 watchdogs, where each red dot (resp. black dot) represents the watchdog (resp. sensor node), and each red circle represents the communication range of a watchdog.

**Figure 4 sensors-23-05943-f004:**
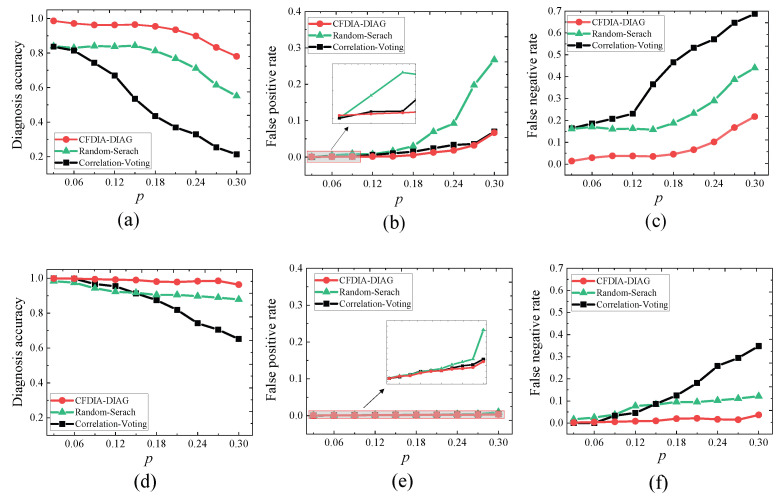
Effectiveness of the three diagnosis algorithms in experiment 1. (**a**) Diagnosis accuracy in the CFDIA situation. (**b**) False positive rate in the CFDIA situation. (**c**) False negative rate in the CFDIA situation. (**d**) Diagnosis accuracy in the SFDIA situation. (**e**) False positive rate in the SFDIA situation. (**f**) False negative rate in the SFDIA situation.

**Figure 5 sensors-23-05943-f005:**
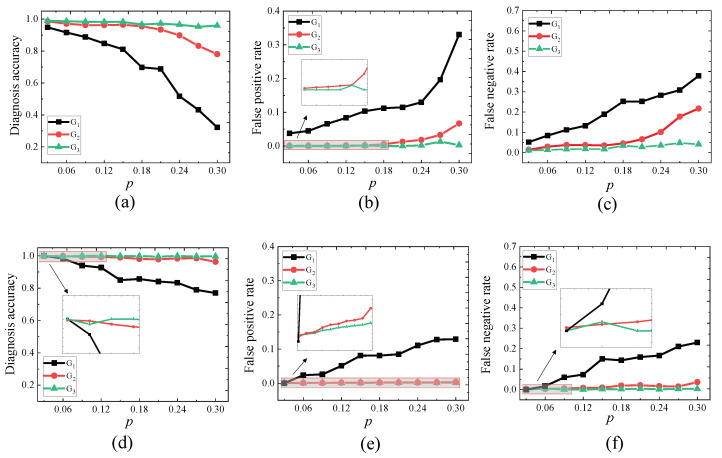
Effectiveness of the CFDIA-DIAG diagnosis algorithms in experiment 2. (**a**) Diagnosis accuracy in the CFDIA situation. (**b**) False positive rate in the CFDIA situation. (**c**) False negative rate in the CFDIA situation. (**d**) Diagnosis accuracy in the SFDIA situation. (**e**) False positive rate in the SFDIA situation. (**f**) False negative rate in the SFDIA situation.

**Figure 6 sensors-23-05943-f006:**
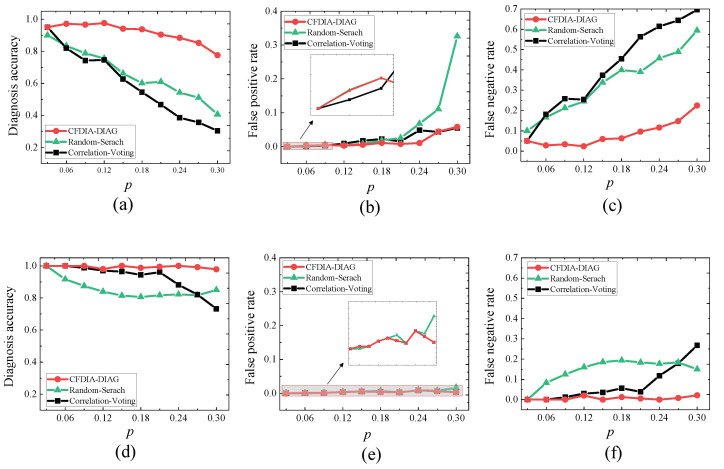
Effectiveness of the three diagnosis algorithms in experiment 3. (**a**) Diagnosis accuracy in the CFDIA situation. (**b**) False positive rate in the CFDIA situation. (**c**) False negative rate in the CFDIA situation. (**d**) Diagnosis accuracy in the SFDIA situation. (**e**) False positive rate in the SFDIA situation. (**f**) False negative rate in the SFDIA situation.

## Data Availability

Data are available upon request.

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
