# Peer review of "A Novel Diagnosis Scheme against Collusive False Data Injection Attack"

_sensors, 2023, doi:10.3390/s23135943_

Round 1

Reviewer 1 Report

The authors study an interesting problem of “maliciously misleading” a WSN by injecting false data on multiple nodes. Reading the paper, the approach seems mathematically sound, and I see the development of stronger mathematical evaluations as an important contribution. 

Yet, in the modelling a number of strong assumptions are made that may prevent too broad generalization of the conclusions. A main challenge can be a detection of a malicious attack versus a natural deviation. A natural deviation can come 1) as a statistical fluctuation of an otherwise stationary process, 2) as time variations (drift) of the statistics over time or 3) can come as a disruptive event. It seems to me that the paper only considers the first possibility.   

The system needs to make a good trade off between false positives and false negatives. Indeed, the author report accuracy and false positives, separately in various curves. I see the paper as a good step in introducing solid mathematical evaluation to address this problem

The authors use synthetic data, which follows their model assumptions. In my perception this is OK, to test the theory. But it comes with a downside, that may have to acknowledged more clearly: The data behaves well, and follows the description of the model. But is it capable to distinguish between an attack and a deviation of class 2) or 3)? I can accept that in a first paper, the algorithms are tested against what these are designed for (to distinguish between an attack versus class 1). 

The paper describes “anomaly detection” in Section 2.2. This suggests that the paper can distinguish between “real anomalies or disruptions in the process” (e.g., a fire starting, a power failure) and “attacks”. In my perception, that is NOT in the scope of the paper. In the conclusions and in the abstract, I recommend a sharper description of this limitation. If I understand the model choices correctly (and the use linear models of autoregressive models and PCA) the newly proposed schemes can monitor attacks from nicely behaving processes but can not distinguish between attacks and process disruptions. As an example, it may work to monitor temperatures gradients in growing plants in horticulture governed by well-functioning heat flow and ventilation, but it cannot be simultaneously used to detect the break-out of a fire or the breaking of a glass wall suddenly exposes the system to outdoor climate conditions. 

If so,  I even recommend to include a limiting statement such as “…. monitoring a stationary process”. The claim:” this paper proposes an algorithm for diagnosing the abnormal sensors in the WSN based on the collection of the consistency outcomes, and validates the effectiveness of the diagnosis algorithm through simulation experiments.” May require some further disclaimer that it is not yet designed to distinguish between “natural abnormal deviating from a Widesense Stationary jointly Gaussian processes” and “attack”.

As an open question:

In an attack-free world, could the new algorithms be used to detect major (unintentional) disruptions, such as a the start of a fire?

Mostly OK, no specific comments

Reviewer 2 Report

The manuscript addresses collusive false data injection attack (CFDIA) in WSN and proposes a scheme to detect the compromised sensors based their consistency. The evaluations show superiority of the proposal over existing schemes. The contributions of this work are enough and the evaluations are comprehensive. My major concerns are as follows:

-         1.  The Introduction section lacks to well highlight the research gap and motivate this work.

-          2. The whole parts of section 5.3 are in italic, which is strange to me.

-         3.  The proposed scheme is compared with Trust-Voting algorithm introduced in 2014. I am wondering whether there is no better algorithm after about 9 years.

Can be improved

Round 2

Reviewer 2 Report

All my concerns have been addressed in the revision.